# Intrinsic Biologically Plausible Adversarial Robustness

## Abstract

Artificial Neural Networks (ANNs) trained with Backpropagation (BP) excel in different daily tasks but have a dangerous vulnerability: inputs with small targeted perturbations, also known as adversarial samples, can drastically disrupt their performance. Adversarial training, a technique in which the training dataset is augmented with exemplary adversarial samples, is proven to mitigate this problem but comes at a high computational cost. In contrast to ANNs, humans are not susceptible to misclassifying these same adversarial samples. Thus, one can postulate that biologically-plausible trained ANNs might be more robust against adversarial attacks. In this work, we chose the biologically-plausible learning algorithm *Present the Error to Perturb the Input To modulate Activity* (PEPITA) as a case study and investigated this question through a comparative analysis with BP-trained ANNs on various computer vision tasks. We observe that PEPITA has a higher intrinsic adversarial robustness and, when adversarially trained, also has a more favorable natural-vs-adversarial performance trade-off. In particular, for the same natural accuracies on the MNIST task, PEPITA's adversarial accuracies decrease on average only by 0.26% while BP's decrease by 8.05%.

## 1 Introduction

Adversarial attacks produce adversarial samples, a concept first described by Szegedy et al. (2014), where the input stimulus is changed by small perturbations that can trick a trained ANN into misclassification. Namely, state-of-the-art Artificial Neural Networks (ANNs) trained with Backpropagation (BP) (Rumelhart et al., 1986; Werbos, 1982) are vulnerable to adversarial attacks (Madry et al., 2019). Although this phenomenon was first observed in the context of image classification (Szegedy et al., 2014; Goodfellow et al., 2015), it has since been observed in several other tasks such as natural language processing (Zhang et al., 2020; Morris et al., 2020), audio processing (Takahashi et al., 2021; Hussain et al., 2021), and deep reinforcement learning (Gleave et al., 2020; Pattanaik et al., 2018). Nowadays, making real-world decisions based on the suggestions provided by ANNs has become an integral part of our daily lives. Hence, these models' vulnerability to adversarial attacks severely threatens the safe deployment of artificial intelligence in everyday-life applications (Sarker, 2021; Akhtar & Mian, 2018). For instance, in real-world autonomous driving, adversarial attacks have been successful in deceiving road sign recognition systems (Eykholt et al., 2018). Researchers have proposed several solutions to address this problem, and adversarial training emerged as the state-of-the-art approach (Wang et al., 2022). In adversarial training, the original dataset, consisting of pairs of input samples with their respective ground-truth labels, is augmented with adversarial data, where the original ground-truth labels are paired with adversarial samples. This additional training data allows the model to correctly classify adversarial samples as well (Goodfellow et al., 2015; Madry et al., 2019). Although adversarial training increases the networks' robustness to adversarial attacks, generating numerous training adversarial samples is computationally costly. To reduce this additional computational burden, researchers have developed new methods for generating adversarial samples more efficiently (Kaufmann et al., 2022; Addepalli et al., 2022; Zheng et al., 2020; Sriramanan et al., 2021). For example, weak adversarial samples created with the Fast Gradient Sign Method (FGSM), which are easy to compute, are used for fast adversarial training (Goodfellow et al., 2015). However, if trained with fast adversarial training, the model remains susceptible to stronger computationally-heavy adversarial attacks, such as the Projected Gradient Descent (PGD) (Kurakin et al., 2016). In this case, the model trained with fast adversarial training can correctly

classify FGSM adversarial samples, but its performance drops significantly (sometimes to zero in the case of "catastrophic overfitting") for PGD adversarial samples (Wong et al., 2020). Several adjustments have been proposed (Kim et al., 2020; Wong et al., 2020; Golgooni et al., 2021; Kang & Moosavi-Dezfooli, 2021) to circumvent this problem and make fast adversarial training more effective, yet there is room for improvement, and this remains still an active area of research. Another caveat to consider when using adversarial training is the trade-off between natural performance (classification accuracy of unperturbed samples) and adversarial performance (classification accuracy of perturbed samples) (Tsipras et al., 2019; Zhang et al., 2019; Moayeri et al., 2022). This natural-vs-adversarial performance trade-off is a consequence of the fact that while the naturally trained models focus on highly predictive features that may not be robust to adversarial attacks, the adversarially trained models select instead for robust features that may not be highly predictive (Zheng et al., 2020).

While these adversarial attacks can easily trick ANNs into misclassification, they appear ineffective for humans (Zhou & Firestone, 2019). Given the disparity between BP's learning algorithm and biological learning mechanisms (Crick, 1989; Grossberg, 1987; Lillicrap et al., 2020), a fundamental research question is whether models trained with biologically-plausible learning algorithms are more robust to adversarial attacks. Over the recent years, due to the importance of researching biologically-inspired learning mechanisms as alternatives to BP, numerous such learning algorithms have been introduced (Lillicrap et al., 2016; Lee et al., 2015; Whittington & Bogacz, 2017; Scellier & Bengio, 2017; Sacramento et al., 2018; Akrout et al., 2019; Meulemans et al., 2021; Hinton, 2022; Bohnstingl et al., 2022; Dellaferrera & Kreiman, 2022). Thus, we investigate in detail for the first time whether biologically-inspired learning algorithms are robust against adversarial attacks. In this work, we chose *Present the Error to Perturb the Input To modulate Activity* (PEPITA), a recently proposed biologically-plausible learning algorithm (Dellaferrera & Kreiman, 2022), as a study case. In particular, we compare BP and PEPITA's learning algorithms in the following aspects:

- their intrinsic adversarial robustness (i.e., when trained solely on natural samples);

- their natural-vs-adversarial performance trade-off when trained with adversarial training;

- and their adversarial robustness against strong adversarial attacks when trained with weak adversarial samples (i.e., samples generated by FGSM).

With this comparison, we open the door to drawing inspiration from biologically-plausible learning algorithms to develop more adversarially robust models.

## 2 Background - PEPITA

PEPITA is a learning algorithm developed as a biologically-inspired alternative to BP (Dellaferrera & Kreiman, 2022). Its core difference from BP is that it does not require a separate backward pass to compute the gradients used to update the trainable parameters. Instead, the computation of the learning signals relies on the introduced second forward pass – see left half of Figure 1. In BP, the network processes the inputs $\mathbf{x} = \mathbf{h}_0$ with one forward pass where $\mathbf{h}_i = \sigma_i(W_i\mathbf{h}_{i-1}), i = 1, \cdots, L$ (indicated with black arrows) to produce the outputs $\mathbf{h}_L$, given $\sigma_i$ the activation function of layer $i$. The network outputs are then compared to the target outputs $\mathbf{y}^*$ through a loss function, $\mathcal{L}$. The error signal $\mathbf{e}$ computed by $\mathcal{L}$ is then backpropagated through the entire network and used to train its parameters (indicated with red arrows). In PEPITA, as seen in the right half of Figure 1, the first forward pass is identical to BP. However, unlike BP, PEPITA feeds the error signals directly to the input layer via a fixed random feedback projection matrix, $F$. That is, the error signal, $\mathbf{e}$, is added to the original input $\mathbf{x}$, producing the modulated inputs $\mathbf{x} + F\mathbf{e}$ which are used for the second forward pass (illustrated with orange arrows).

In PEPITA, the gradients used to update the synaptic weights are then computed through the difference between the activations of the neurons in the first and second forward passes. If the network's output matches the target for the first forward pass, the error signal $\mathbf{e}$ would be zero. Thus, there would not be any synaptic weight updates because the neural activations in the second forward pass would not differ from the first forward pass. On the contrary, if the network made a wrong prediction in the first pass, then the

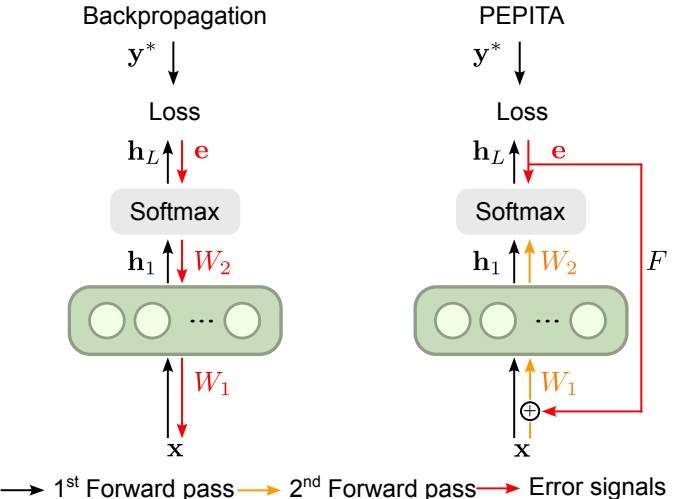

Figure 1: **Comparison of BP and PEPITA.** Schematic of BP's and PEPITA's architectures and learning mechanics with a single hidden layer.

---

**Algorithm 1** PEPITA's Training Algorithm

---
**Given:** input $\mathbf{x}$, label $\mathbf{y}^*$, and an activation function for each layer $\sigma_i(\cdot)$
\# Standard forward pass
$\mathbf{h}_0 = \mathbf{x}$
**for** $i = 1$ **to** $L$ **do**
    $\mathbf{h}_i = \sigma_i(W_i\mathbf{h}_{i-1})$
    $\mathbf{e} = \mathbf{h}_L - \mathbf{y}^*$
**end for**
\# Modulated forward pass
$\mathbf{h}_0^{mod} = \mathbf{x} + F\mathbf{e}$
**for** $i = 1$ **to** $L$ **do**
    $\mathbf{h}_i^{mod} = \sigma_i(W_i\mathbf{h}_{i-1}^{mod})$
    **if** $i < L$ **then**
        $\Delta W_i = (\mathbf{h}_i - \mathbf{h}_i^{mod}) \cdot (\mathbf{h}_{i-1}^{mod})^T$
    **else**
        $\Delta W_L = \mathbf{e} \cdot (\mathbf{h}_{L-1}^{mod})^T$
    **end if**
    \# Update forward parameters with $\Delta W_i$ and an optimizer of choice
**end for**

---

neuronal activations in the second forward pass would differ, and the synaptic weights would be updated to reduce the mismatch. Note that for the last layer, the teaching signal is the error itself, which is why $\mathbf{e}$ is also fed directly to the output layer.

With this learning scheme, PEPITA sidesteps the biologically-implausible requirement of BP to backpropagate gradient information through the entire network hierarchical architecture, allowing the training of the synaptic weights to be based solely on spatially local information through a two-factor Hebbian-like learning rule. Therefore, while BP uses exact gradients for learning, that is, the exact derivative of the loss function with respect to its trainable parameters, PEPITA uses a very different learning mechanism that results in approximates of BP's exact gradients.

To illustrate this difference, we write the explicit gradient weight updates for BP and PEPITA for the network represented in Figure 1. For the output layer, $W_2 = W_L$, the synaptic weight updates of BP are

computed as follows:

$$\Delta W_L^{BP} = \frac{\partial \mathcal{L}}{\partial W_L} = \frac{\partial \mathcal{L}}{\partial \mathbf{h}_L} \cdot \frac{\partial \mathbf{h}_L}{\partial W_L} \tag{1}$$

$$= \mathbf{e} \cdot \left( \sigma_L'(\mathbf{z}_L) \cdot (\mathbf{h}_{L-1})^T \right), \tag{2}$$

where $\mathbf{z}_L = W_L \mathbf{h}_{L-1}$ and $\mathbf{e} = \partial \mathcal{L}/\partial \mathbf{h}_L$. For a simple mean-squared-error function (MSE) we have that $\mathbf{e} = \mathbf{h}_L - \mathbf{y}^*$. For the output layer weight updates, PEPITA leverages the same error signal and update structure as BP, the difference lying only on the scaling factor $\mathbf{z}_L$ and the pre-synaptic activation $\mathbf{h}_{L-1}^{mod}$ being given by the modulated forward pass:

$$\Delta W_L^{PEPITA} = \mathbf{e} \cdot (\mathbf{h}_{L-1}^{mod})^T. \tag{3}$$

Hence, for the output layer, we have that $\Delta W_L^{PEPITA} \approx \Delta W_L^{BP}$, since the derivative of the activation function is bounded, and the difference between $\mathbf{h}_{L-1}^{mod}$ and $\mathbf{h}_{L-1}$ is small because the feedback projection matrix $F$ is scaled by a small constant, which guarantees that the perturbations added to the input are small (Dellaferrera & Kreiman, 2022). On the contrary, considering the case with one hidden layer as illustrated in Figure 1, the synaptic weight updates of the hidden layer will differ more significantly since we have:

$$\Delta W_1^{BP} = \frac{\partial \mathcal{L}}{\partial W_1} = \frac{\partial \mathcal{L}}{\partial \mathbf{z}_L} \frac{\partial \mathbf{z}_L}{\partial \mathbf{z}_1} \frac{\partial \mathbf{z}_1}{\partial W_1} = \boldsymbol{\delta}_1 \cdot \mathbf{x}^T \tag{4}$$

$$\Delta W_1^{PEPITA} = (\mathbf{h}_1 - \mathbf{h}_1^{mod}) \cdot (\mathbf{x}^{mod})^T, \tag{5}$$

where $\boldsymbol{\delta}_1 = (W_2^T \boldsymbol{\delta}_2) \cdot \sigma_1'(\mathbf{z}_1)$ and $\boldsymbol{\delta}_2 = \mathbf{e}$. In general, we can write $\Delta W_l^{BP} = \boldsymbol{\delta}_l \cdot \mathbf{h}_{l-1}^T$, where $\boldsymbol{\delta}_l$ is recursively obtained by writing it w.r.t. $\boldsymbol{\delta}_{l+1}$: $\boldsymbol{\delta}_l = (W_{l+1}^T \boldsymbol{\delta}_{l+1}) \cdot \sigma_l'(\mathbf{z}_l)$.

Like BP, the FGSM and PGD adversarial attacks rely on backpropagating the exact derivatives of the loss function but all the way back to the input samples instead of just to the first hidden layer. For instance, the FGSM attack involves perturbing the input stimulus in the direction of the gradient of the loss function with respect to the input, being that a small $\epsilon$ constant typically scales the added perturbation. An adversarial sample can computed as follows:

$$\mathbf{x}_{\text{adv}} = \mathbf{x} + \epsilon \cdot \text{sign}\left( \frac{\partial \mathcal{L}}{\partial \mathbf{x}} \right) \tag{6}$$

$$= \mathbf{x} + \epsilon \cdot \text{sign}\left( \frac{\partial \mathcal{L}}{\partial \mathbf{z}_L} \frac{\partial \mathbf{z}_L}{\partial \mathbf{z}_1} \frac{\partial \mathbf{z}_1}{\partial \mathbf{x}} \right) \tag{7}$$

$$= \mathbf{x} + \epsilon \cdot \text{sign}\left( W_1^T \boldsymbol{\delta}_1 \right), \tag{8}$$

where $\mathbf{x}_{\text{adv}}$ is the adversarial example. Comparing Equations equation 8 and equation 4, we see that the driving signal, $\boldsymbol{\delta}_1$, is the same. Thus, we postulate that adversarial attacks have a more significant impact on BP-trained networks. The PGD attack follows a similar gradient path, applying FGSM iteratively with small stepsizes while projecting the perturbed input back into an $\epsilon$ ball. As PEPITA-trained models do not use these exact derivatives for learning, they form excellent candidates to be explored in the context of adversarial robustness. In the case of PEPITA, the attacker will still use the transposed feedforward pathway to compute the adversarial samples. Otherwise, if the random feedback projection matrix $F$ is used, the adversarial samples produced would be too weak, and the model could easily classify them correctly (Akrout, 2019).

## 3 Results

### 3.1 Model training details

For our comparative study, we used four benchmark computer vision datasets: MNIST (LeCun, 1998), Fashion-MNIST (F-MNIST) (Xiao et al., 2017), CIFAR-10 and CIFAR-100 (Krizhevsky et al., 2014). For

both BP and PEPITA, we used the same network architectures and training schemes described in Dellaferrera & Kreiman (2022). However, we added a bias to the hidden layer, as we observed a substantial performance improvement. The learning rule for this bias is similar to the one for the synaptic weights, but the pre-synaptic activation is fixed to one, i.e., $\mathbf{h}_{i-1}^{mod} \coloneqq \mathbf{1}$. Thus, similarly to the update rule for the synaptic weights (see Algorithm 1), the bias update rule can be written as $\Delta \mathbf{b}_i = (\mathbf{h}_i - \mathbf{h}_i^{mod})$ for $i < L$ and $\Delta \mathbf{b}_L = \mathbf{e}$. The network architecture consists of a single fully connected hidden layer with 1024 ReLU neurons and a softmax output layer (as represented in Figure 1). For all the tasks, we used: the MSE loss, 100 training epochs with early stopping, and the momentum Stochastic Gradient Descent (SGD) optimizer (Qian, 1999). Furthermore, we used a mini-batch size of 64, neuronal dropout of 10%, weight decay at epochs 60 and 90 with a rate of 0.1, and the He uniform initialization (He et al., 2015b) with the feedback projection matrix $F$ initialization scaled by 0.05. [1]

We optimized the learning rate hyperparameter, $\eta$, by performing a grid search across 50 different values with $\eta \in [0.001, 0.3]$. All the values presented in the result tables of this work can be reproduced using the configurations available in the shared public repository. We defined the best-performing model as the model with the best natural accuracy on the validation dataset, consisting of natural samples the model has not yet seen. We chose this model selection criterion because, in real-world applications, the networks' natural performance is most important to the user, and adversarial samples are considered outside of the norm. Thus, unless stated otherwise, we do not select the models based on the best adversarial validation accuracy, as we found that this comes at the cost of significantly worse natural performance. The values reported throughout this section are the mean $\pm$ standard deviation of the test accuracy for 5 different random seeds. The adversarial attacks were done using the open-source library *advertorch.attacks* (Ding et al., 2019), which follows the original implementations of FGSM and PGD, as introduced in (Goodfellow et al., 2015) and (Kurakin et al., 2016), respectively. We defined an attack step size of 0.1 and a maximum distortion bound of 0.3 to create the FGSM and PGD adversarial samples and used 40 iterations for PGD. Note that the maximum and minimum pixel values of the adversarial images are the same as those of the original natural images.

## 3.2 Baseline natural and adversarial performance

Table 1 shows the models' natural and adversarial performances when trained naturally, i.e., without adversarial samples in the training dataset, and with natural validation accuracy as the hyperparameter selection criterion. In line with the results reported in the literature (Dellaferrera & Kreiman, 2022), PEPITA achieves a lower natural performance than BP. Notably, neither model is robust to adversarial attacks since neither has been adversarially trained nor has their hyperparameter selection criterion set to value adversarial robustness as an advantage.

Hyperparameter selection criterion: natural validation accuracy

| Train | Test Data | MNIST [%] | F-MNIST [%] | CIFAR-10 [%] | CIFAR-100 [%] |
|---|---|---|---|---|---|
| BP | natural | $98.58^{\pm 0.05}$ | $90.52^{\pm 0.03}$ | $57.05^{\pm 0.35}$ | $27.54^{\pm 0.25}$ |
| (w/o adv samples) | PGD | $2.504^{\pm 0.48}$ | $1.961^{\pm 0.46}$ | $0.000^{\pm 0.00}$ | $0.00^{\pm 0.00}$ |
| PEPITA | natural | $98.16^{\pm 0.04}$ | $86.46^{\pm 0.66}$ | $52.15^{\pm 0.25}$ | $25.88^{\pm 0.26}$ |
| (w/o adv samples) | PGD | $0.056^{\pm 0.31}$ | $0.00^{\pm 0.00}$ | $0.00^{\pm 0.00}$ | $0.00^{\pm 0.00}$ |

Table 1: Natural test accuracy and adversarial test accuracy with the PGD attack for 5 different random seeds. Here, the model is trained without adversarial samples, and the hyperparameter selection criterion is the natural validation accuracy. With the following order {MNIST, F-MNIST, CIFAR-10, CIFAR-100}: $\eta^{\text{BP}} = \{0.123, 0.051, 0.008, 0.035\}$ and $\eta^{\text{PEPITA}} = \{0.255, 0.016, 0.012, 0.029\}$.

---

[1] PyTorch implementation of all methods will be available at a public repository.

### 3.3 PEPITA's higher intrinsic adversarial robustness

When using the same training procedure as in the section above (natural training) but selecting the hyperparameter search criterion to value accuracy on adversarial validation samples, PEPITA shows a higher intrinsic adversarial robustness compared to BP. This manifests itself in a significantly lower drop in performance when comparing the natural and the adversarial performance for BP and PEPITA, comparing rows 1 and 2 as well as rows 3 and 4 in Table 2. Although this model selection criterion leads to a higher level of adversarial robustness, one can observe a natural-vs-adversarial performance trade-off. For instance, compare row 1 of Table 2) with row 1 of Table 1 for BP and row 3 of Table 2) with row 3 of Table 1 for PEPITA. However, we can conclude that PEPITA is significantly more robust against adversarial as it exhibits a much more favorable trade-off across all datasets.

Furthermore, BP-trained models cannot become adversarially robust with more complex tasks like Fashion-MNIST, CIFAR-10, and CIFAR-100. During the hyperparameter search of BP, it was observed that the learning rates tended to be much larger with the current selection criterion (best accuracy on adversarial validation samples). Consequently, the models either did not converge during learning or they did not learn at all. In the first case, the results were highly variable (see BP Fashion-MNIST results in Table 2), and in the second case, the natural and adversarial performances became almost random (see BP CIFAR-10 results in Table 2).

Hyperparameter selection criterion: adversarial validation accuracy

| Train | Test Data | MNIST [%] | F-MNIST [%] | CIFAR-10 [%] | CIFAR-100 [%] |
|---|---|---|---|---|---|
| BP | natural | $94.22^{\pm 0.40}$ | $44^{\pm 34}$ | $10.00^{\pm 0.04}$ | $9.08^{\pm 0.33}$ |
| (w/o adv samples) | PGD | $92.72^{\pm 0.36}$ | $23^{\pm 11}$ | $9.98^{\pm 0.05}$ | $0.33^{\pm 0.24}$ |
| PEPITA | natural | $97.69^{\pm 0.16}$ | $80.65^{\pm 0.74}$ | $41.82^{\pm 1.57}$ | $17.10^{\pm 0.72}$ |
| (w/o adv samples) | PGD | $97.56^{\pm 0.18}$ | $80.48^{\pm 0.73}$ | $41.73^{\pm 1.49}$ | $16.76^{\pm 0.65}$ |

Table 2: Natural test accuracy and adversarial test accuracy with the PGD attack for 5 different random seeds. Here, the model is trained without adversarial samples, and the hyperparameter selection criterion is the adversarial validation accuracy. With the following order {MNIST, F-MNIST, CIFAR-10, CIFAR-100}: $\eta^{\text{BP}} = \{0.378, 0.273, 0.039, 0.180\}$ and $\eta^{\text{PEPITA}} = \{0.378, 0.037, 0.025, 0.061\}$.

### 3.4 PEPITA's advantageous adversarial training

As the next step, we investigate how advantageous adversarial training is for BP and PEPITA, i.e., how robust to adversarial attacks the models can become when their training dataset is augmented with adversarial samples. We set the hyperparameter search selection criterion to be the natural validation accuracy, the same as in Section 3.2, and observe that the resulting trained models are now robust against adversarial attacks (compare Tables 1 and 3). In particular, in Table 3, we see that PEPITA achieves a better adversarial testing performance and less natural performance degradation compared to BP, except for CIFAR-100 where both models are not significantly adversarially robust. Moreover, for the MNIST and Fashion-MNIST datasets, BP has better natural test accuracies. Hence, although Table 3 suggests that PEPITA offers a better natural-vs-adversarial performance trade-off, comparing the adversarial robustness between BP and PEPITA becomes difficult for these datasets.

To better understand this trade-off, we chose 13 different natural accuracy values distributed between 96% and 99% and selected the BP-trained and PEPITA-trained models with the closest natural accuracy to these values and best adversarial performance during the hyperparameter selection. Because finding models that perform well adversarially while maintaining high natural performance is challenging, only the 13 different learning rates (still inside the interval [0.001, 0.3] as done for all the experiments) found that lead to models with natural performances within the desired range and good adversarial performances were included. Then, we trained these models for 5 different random seeds and averaged over their natural and adversarial accuracies. We plotted these results in Figure 2(A), which shows that PEPITA performs significantly better than BP for similar values of natural performance. In particular, the average decrease in

adversarial performance for the same values of natural performance, that is, the average value of (natural [%] - PGD [%]) across the 13 plotted points, with each point being the average performance of the trained model over 5 different random seeds, is 0.26% for PEPITA and 8.05% for BP. Moreover, we verified that even if we double the number of training epochs for BP, its natural and adversarial accuracies remain approximately the same, indicating that the model has converged in its learning dynamics – see Figure 2(B). Hence, even after extensive hyperparameter searches and increased training epochs, we could not find BP-trained models with a better natural-vs-adversarial performance trade-off.

Hyperparameter selection criterion: natural validation accuracy

| Train | Test Data | MNIST [%] | F-MNIST [%] | CIFAR-10 [%] | CIFAR-100 [%] |
|---|---|---|---|---|---|
| BP | natural | $98.73^{\pm 0.06}$ | $85.16^{\pm 0.17}$ | $35.83^{\pm 0.37}$ | $12.45^{\pm 0.38}$ |
| (w/ PGD adv samples) | PGD | $89.93^{\pm 0.03}$ | $67.42^{\pm 0.21}$ | $8.58^{\pm 0.16}$ | $2.11^{\pm 0.15}$ |
| PEPITA | natural | $98.18^{\pm 0.10}$ | $83.73^{\pm 0.76}$ | $45.12^{\pm 0.89}$ | $22.30^{\pm 0.16}$ |
| (w/ PGD adv samples) | PGD | $97.30^{\pm 0.41}$ | $83.19^{\pm 0.68}$ | $44.94^{\pm 0.83}$ | $2.9^{\pm 1.7}$ |

Table 3: Natural test accuracy and adversarial test accuracy with the PGD attack for 5 different random seeds. Here, the models are adversarially trained (i.e., trained with a dataset augmented by the adversarial samples) with PGD adversarial samples. The hyperparameter selection criterion is the natural validation accuracy. With the following order {MNIST, F-MNIST, CIFAR-10, CIFAR-100}: $\eta^{\mathrm{BP}} = \{0.052, 0.030, 0.012, 0.014\}$ and $\eta^{\mathrm{PEPITA}} = \{0.067, 0.012, 0.012, 0.021\}$.

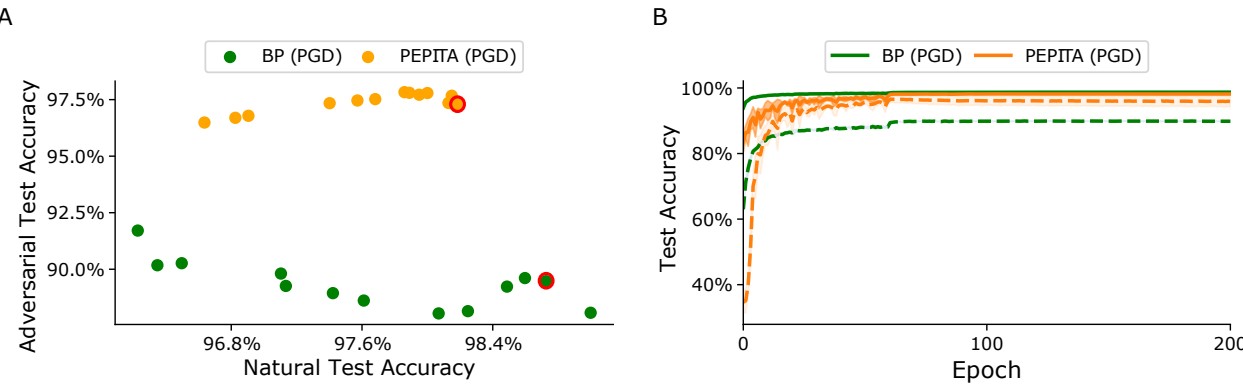

Figure 2: **PEPITA's advantageous adversarial training.** The results presented here are for BP and PEPITA models trained adversarially with PGD samples on the MNIST task for 5 different random seeds. (A) Natural-vs-adversarial performance trade-off: the most adversarially robust models were selected for different natural accuracy values. That is, different natural accuracy values distributed between 96% and 99% were chosen, and the models with the closest natural accuracy to these values and best adversarial performance were selected during the hyperparameter selection. Each data point's coordinates stand for the average performances over the 5 different random seeds, that is, the axes in the plot represent the adversarial and natural average test accuracies across these random seeds. The values reported in the first column (MNIST) of Table 3, which correspond to the models with the best adversarial accuracies, are marked in red. (B) Natural (represented by the full lines) and adversarial (represented by the dashed lines) test accuracies of the models encircled in (A). To demonstrate that the performance of BP does not increase further, we trained both models for twice the amount of epochs. The shaded area represents the standard deviation across the models trained with 5 different random seeds.

### 3.5 PEPITA's advantageous fast adversarial training

After demonstrating PEPITA's intrinsic adversarial robustness and beneficial natural-vs-adversarial performance trade-off, we now investigate PEPITA's capabilities in fast adversarial training (Goodfellow et al., 2015). Table 4 reports the results obtained when using fast adversarial training, i.e., when augmenting the

training dataset with FGSM adversarial samples, which are fast to compute, while evaluating the model with the more sophisticated PGD attack, whose samples are slower to compute. Similar to Sections 3.2 and 3.4, we set the hyperparameter selection criterion to be the natural validation accuracy. We observe that when attacking the trained models with strong attacks, such as with PGD adversarial samples, the decrease in adversarial performance is much less significant for PEPITA than for BP, indicating that the PEPITA-trained models can generalize better from the FGSM samples. This is an important property for two reasons: firstly, the FGSM samples are faster to compute and, thus, reduce the computational overhead for training, and secondly, this better generalization can be beneficial in cases where the attack method during testing is unknown during training time. Therefore, PEPITA is in an advantageous position for this type of scenario compared to BP. Moreover, neither BP nor PEPITA-trained models suffer from catastrophic overfitting for this specific network architecture since the PGD testing accuracies do not drop to zero. According to Zhu et al. (2022), this is the case for two reasons: first, our network is wide (1024 neurons) and in the over-parameterized regime, i.e., the network has more trainable parameters than there are samples in the dataset, which increases adversarial robustness; and second, we use He weight initialization (He et al., 2015a) with a shallow network (a single hidden layer), which prevents a decrease in adversarial robustness.

Hyperparameter selection criterion: natural validation accuracy

| Train | Test Data | MNIST [%] | F-MNIST [%] | CIFAR-10 [%] | CIFAR-100 [%] |
|---|---|---|---|---|---|
| BP (w/ FGSM adv samples) | natural | $98.93^{\pm0.05}$ | $84.90^{\pm0.03}$ | $51.56^{\pm0.43}$ | $26.59^{\pm0.08}$ |
| | FGSM | $91.04^{\pm0.13}$ | $66.31^{\pm0.25}$ | $45.06^{\pm3.38}$ | $2.51^{\pm0.37}$ |
| | PGD | $86.25^{\pm0.09}$ | $57.95^{\pm0.33}$ | $0.05^{\pm0.04}$ | $1.19^{\pm0.08}$ |
| PEPITA (w/ FGSM adv samples) | natural | $98.00^{\pm0.14}$ | $80.70^{\pm0.95}$ | $41.22^{\pm2.01}$ | $17.89^{\pm0.52}$ |
| | FGSM | $97.91^{\pm0.13}$ | $80.68^{\pm0.96}$ | $41.22^{\pm2.22}$ | $17.68^{\pm0.44}$ |
| | PGD | $97.81^{\pm0.12}$ | $80.27^{\pm1.05}$ | $41.00^{\pm2.20}$ | $17.53^{\pm0.48}$ |

Table 4: Natural test accuracy and adversarial test accuracies with the PGD and FGSM attacks for 5 different random seeds. Here, the models are adversarially trained with FGSM adversarial samples. The hyperparameter selection criterion is the natural validation accuracy. With the following order {MNIST, F-MNIST, CIFAR-10, CIFAR-100}: $\eta^{\mathrm{BP}} = \{0.097, 0.010, 0.012, 0.027\}$ and $\eta^{\mathrm{PEPITA}} = \{0.097, 0.027, 0.016, 0.041\}$.

## 4 Discussion

Our paper demonstrates for the first time that biologically-inspired learning algorithms can lead to ANNs that are more robust against adversarial attacks than BP. We found that, unlike BP, PEPITA-trained models can be intrinsically robust against adversarial attacks. That is, naturally trained (i.e., only using natural samples) PEPITA models can be adversarial robust without the computationally heavy burden of adversarial training. A similar finding of intrinsic adversarial robustness has been demonstrated by Akrout (2019) for the biologically-plausible learning algorithm Feedback Alignment (FA) (Lillicrap et al., 2016). However, in this previous work Akrout (2019), a non-common practice that leads to much weaker adversarial attacks was used: the attackers use the FA's random feedback matrices to generate adversarial samples instead of the transposed feedforward pathway. Hence, their analysis in Akrout (2019) differs from our approach, where we let the attacker fully access the network architecture and synaptic weights and craft its adversarial samples through the transposed forward pathway. Moreover, we found that PEPITA does not suffer from the natural-vs-adversarial performance trade-off as severely as BP, as its models can be more adversarially robust than BP while losing less natural performance. Lastly, we found that PEPITA benefits much more from fast adversarial training than BP, i.e., when trained with samples generated from weaker adversarial attacks, it reports much better adversarial robustness against strong attacks.

Through this study, we postulate the advantage of using alternative feedback pathways, as many biologically-plausible learning algorithms do, to enhance adversarial robustness. The feedback pathway serves as the mechanism by which the teaching signal calculated at the output level is transmitted throughout the network. This signal is then used to adjust the synaptic weights of the network. While for BP, the feedback

pathway is simply the transposed of the forward pathway, for many alternative biologically-plausible learning algorithms, the feedback pathway is separate from the forward weights. Since classic gradient-based adversarial attacks leverage the transposed forward pathway to generate targeted perturbations of the input stimulus, updating synaptic weights through alternative feedback routes may bolster robustness against these adversarial perturbations. Concluding, this work opens the door to a variety of research avenues, which are discussed in the subsequent section.

## 4.1 Limitations and future work

Based on the insights gained from this work, we postulate that PEPITA's increased adversarial robustness arises from its alternative feedback pathway. Therefore, a natural next step would be to investigate whether other biologically-plausible learning algorithms with alternative feedback mechanisms exhibit a similar robustness (Lee et al., 2015; Whittington & Bogacz, 2017; Scellier & Bengio, 2017; Sacramento et al., 2018; Meulemans et al., 2021). Moreover, a theoretical understanding of the link between this alternative feedback pathway learning mechanism and adversarial robustness can enable the identification of the exact properties that improve the natural-vs-adversarial performance trade-off, which in turn can be used to specifically develop adversarially robust models. For instance, by analyzing the weight gradients of both PEPITA and traditional BP, it is possible to modify BP's gradients by introducing noise that aligns these gradients more closely with those of PEPITA. This study could be an experimental method to evaluate if the alternative feedback pathway used to compute the approximate gradients employed by PEPITA is crucial to the model's robustness to adversarial attacks. Additionally, various strategies like data augmentation and incorporating both real and synthetic data (Li & Spratling, 2022; Carmon et al., 2019; Wang et al., 2023) offer promising paths to boost adversarial robustness and achieve more favorable balances between natural and adversarial performance. Exploring the potential of these strategies to enhance the effectiveness of biologically-inspired algorithms, like PEPITA, represents an intriguing future research direction. Another limitation in generalizing these results is that our exploration was confined to PGD and FGSM adversarial attacks. There are numerous other methods, including additional white-box and black-box attacks Zhang et al. (2023), that warrant investigation in future studies.

On another aspect, PEPITA has recently been extended to deeper networks (up to five hidden layers) and tested with different parameter initialization schemes (Srinivasan et al., 2023). Thus, studying the impact of these variable characteristics of the model, such as width, depth, and initialization, on PEPITA's adversarial robustness would be beneficial (as done in Zhu et al. (2022)). PEPITA's natural performance has also been recently improved through the learning of the feedback projection matrix (Srinivasan et al., 2023), so it would be interesting to study whether this also improves adversarial robustness.

## 4.2 Conclusion

To conclude, we demonstrated that ANNs trained with PEPITA, a recently proposed biologically-inspired learning algorithm, are more adversarially robust than BP-trained ANNs. In particular, we showed through several computational experiments that PEPITA significantly outperforms BP in an adversarial setting. Thus, we propose that alternative feedback pathways in these algorithms enhance the adversarial robustness of the trained models. Our analysis opens the door to drawing inspiration from biologically-plausible learning algorithms for designing more adversarially robust models. In conclusion, our work contributes to the important cause of developing safer and more trustworthy artificial intelligence systems.

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
