# OpenReview forum: "Intrinsic Biologically-Plausible Adversarial Robustness"
_TMLR — Rejected by TMLR_

### Review · Reviewer_YRKX · 2024-06-28

**Summary Of Contributions:**

The work considers the robustness properties of networks trained with a recently proposed learning algorithm (PEPITA). This learning rule does not need to consider the transposes of weight matrices, instead computes the learning signal by performing two forward passes. The authors considered networks with a single deep layer, which were trained on four distinct benchmarks either using back-propagation (BP) or PEPITA. With a line of experiments that also include training with adversarial samples (computed using two distinct algorithms FGSM and PGD), the authors argue that networks trained with PEPITA show increased inherent robustness to adversarial attacks compared to the networks trained with BP.

**Audience:**

Yes

**Broader Impact Concerns:**

No concerns.

**Claims And Evidence:**

No

**Requested Changes:**

I would like to thank the authors for writing this work, I found it quite enjoyable to read. Yet, unfortunately, I am leaning to reject in its current form. That being said, I am willing to accept if the authors address the major points below and answer my questions, I have also provided some minor comments that I believe would help the authors but may not need to be addressed for an acceptance:

## Major points

- Please provide a proper methods section in the Appendix, which will mitigate the need to look into the codebase to reproduce and understand your work. Please note that though making a reproduction repository public is very welcome, and encouraged, it does not alleviate the need for a proper methods section that is self-sufficient for the full paper. Please explain how each experiment is conducted, in details, in this section.

- "Notably, neither model is robust to adversarial attacks since neither has been adversarially trained nor has their hyperparameter selection criterion set to value adversarial robustness as an advantage." So, this is a bit concerning for me. Combined with later results, it seems that PEPITA becomes adversarially robust only when trained with large learning rates. Could you please report two plots in a figure, with the x-axis being the value of the learning parameter and the y-axis being the natural and adversarial (PGD) accuracies, respectively. It would be great to see how PEPITA and BP perform as a function of learning parameters, and *without* any adversarial training samples. You are welcome to add new plots with adversarial training samples to the supplementary, though not required.

- "As the next step, we investigate how advantageous adversarial training is for BP and PEPITA, i.e., how robust to adversarial attacks the models can become when their training dataset is augmented with adversarial samples." Please state explicitly that these are trained with PGD. Also, PGD based adversarial training is never truly discussed in the paper. Please provide a summary for the readers in a supplementary methods section. This is a general thread in this paper, in which the readers should regularly go back to previous papers to follow some of the arguments. This should NOT be the case for a paper with clear and convincing presentation.

- I personally did not enjoy the several lines of claims to be the "first". They are usually followed by asterisks and additional explanations of how the authors are different from prior work. For example: "A similar finding of intrinsic adversarial robustness has been demonstrated by Akrout (2019) for the biologically-plausible learning algorithm Feedback Alignment (FA) (Lillicrap et al., 2016)." The authors do not need these claims, nor do they seem to be correct. Of course, the authors are first to do this particular research, which is why the work is being considered for publication in the first place. Thus, these claims are either too trivial to be there (after several asterisks), or are ignorant of similar work that the authors later discuss. To me, sentences like "Our paper demonstrates for the first time that biologically-inspired learning algorithms can lead to ANNs that are more robust against adversarial attacks than BP." definitely raised an eye-brow and are NOT necessary nor helping the authors.

## Questions

- "PEPITA uses a very different learning mechanism that results in approximates of BP’s exact gradients." Is the learning rule somehow "approximating" the gradient, which to me implies that in some limit the gradient is recovered? Or, is it just simply a different learning signal? Can the authors please clarify?

- "in the second case, the natural and adversarial performances became almost random" What is the difference between almost random and random? I was confused by the phrasing "almost".

- The fact that high learning rates lead to robustness is very well known in the literature. In the paper, though it is alluded to, this aspect is not properly discussed and relevant literature not cited. Could you please explain why, or am I missing something?

- "Moreover, we verified that even if we double the number of training epochs for BP, its natural and adversarial accuracies remain approximately the same, indicating that the model has converged in its learning dynamics – see Figure 2(B)." This is very interesting and was the point where I was partially convinced. Can you please show the same results with different hyperparameters? For example, what if we used ADAM? What if we used a higher momentum? I would love to see robustness of this result to some simple controls. If not, can you please explain the rationale for these missing controls?

- I do not understand why weight decay was only performed at distinct epochs. Why not apply weight decay constantly? Similar to the learning rates, can you please add a new figure with the weight decay values varied? Overall, weight decay would be another control I would like to see regarding how it affects robustness and I was not able to see a satisfactory answer in this work.

- One of the main claims in the paper is that since the adversarial samples are computed using similar algorithms as BP, BP is more susceptible to them. I would like to see this being explored a bit more (or the claim can be downplayed alternatively, though I hope the authors will choose to show us more). For example, what happens if we get the adversarial samples using a bio-plausible algorithm during training, but still test w.r.t. PGD samples? The authors note in the discussion "However, in this previous work Akrout (2019), a non-common practice that leads to much weaker adversarial attacks was used: the attackers use the FA’s random feedback matrices to generate adversarial samples instead of the transposed feedforward pathway. Hence, their analysis in Akrout (2019) differs from our approach, where we let the attacker fully access the network architecture and synaptic weights and craft its adversarial
samples through the transposed forward pathway." I find this method of training with bio-plausible samples to be more appropriate for the claims the authors are making in this work, since a bio-plausible learning algorithm should also use adversarial samples that are obtained in a bioplausible manner. Is it possible for you to show some new experiments in this direction?

## Minor points

- "In contrast to ANNs, humans are not susceptible to misclassifying these same adversarial samples." As a part-experimental neuroscientist, this sentence was quite surprising for me. We know humans are susceptible to adversarial samples, just not in a way that ANNs are. Think about optical illusions.

- I believe the writing needs to be significantly polished. For example, the introduction is a one full paragraph, which is very hard to read and follow. It would be great if the authors spent some time to ease the reader into the topic with several paragraphs, with each paragraph providing a *single* argument/fact.

- "Because finding models that perform well adversarially while maintaining high natural performance is challenging, only the 13
different learning rates (still inside the interval [0.001, 0.3] as done for all the experiments) found that lead to models with natural performances within the desired range and good adversarial performances were included." I believe this sentence is too long, and needs to be broken into two parts. There are several sentences with similar properties, I recommend the authors perform a proof read and break up very long sentences to allow the reader to more easily catch up with the arguments.

- Can you please add the unity line to Figure 2A? It would be helpful to know where the adversarial accuracies stand with respect to natural accuracies.

- "performance is much less significant for PEPITA than for BP," I do not believe this phrasing is appropriate in a scientific context. Something is either statistically significant or not.

- I appreciated the limitation section. Can you please add the methodological aspects, e.g. limited network structure, only SGD, no proper weight decay etc.

**Strengths And Weaknesses:**

## Strengths

I believe the work satisfies the criteria for interest to TMLR audience. The fresh look into the relationship between the robustness and biologically plausible learning is very welcome. The case study is simple enough that a general audience can follow it and powerful enough that it allows the authors to demonstrate their main claims. Overall, I really liked this paper, though see below for my request for major revisions.

## Weaknesses

I believe the work does not fully satisfy TMLR's criteria for accurate, convincing and clear evidence. The paper is missing a proper methods section. Most details are deferred to a "public" repository, which is not public to the reviewers, and in any case the paper should have been self-sufficient. I believe some control experiments are needed to fully support the claims made in the paper, please see below

---

### Review · Reviewer_qYc9 · 2024-07-03

**Summary Of Contributions:**

This paper investigates the adversarial robustness of models optimized using Present the Error to Perturb the Input To modulate Activity (PEPITA), a biologically-inspired learning algorithm. Focusing on FGSM and PGD attacks on a one-hidden-layer model, a comparative studies demonstrates that PEPITA-trained models offer advantages over backpropagation (BP)-trained models in areas such as intrinsic robustness, the robustness-accuracy trade-off, and generalization to stronger PGD attacks when trained with FGSM. The paper also includes discussions on hypotheses regarding the observed robustness benefits, as well as the limitations and future directions for this line of research.

**Audience:**

Yes

**Claims And Evidence:**

No

**Requested Changes:**

**Related to presentations:**
1. In the abstract, the term **targeted** perturbation is mentioned. This term is used to describe perturbations that cause misclassification to a specific class. However, since this is not the focus of the paper, the term **targeted** should be used with caution.
2. To my knowledge, the term "Fast Adversarial Training" refers to the adversarial training method described in [C], where the perturbation consists of random initialization followed by FGSM. However, in this paper, "fast adversarial training" seems to refer to the method described in [D], where the perturbation is the vanilla FGSM. I believe the author is referring to vanilla FGSM training in the paper. It would be better to clarify the exact method being used or perhaps use a different term.
3. The work by Kurakin et al. is cited when mentioning PGD, which I believe is incorrect. PGD with random initialization was proposed by Madry et al., while Kurakin et al. proposed the iterative FGSM method, often referred to as the Basic Iterative Method.
4. At the end of the first paragraph in the Introduction: "This natural-vs-adversarial ... is a consequence of the fact that ..." This is a very strong claim, and there are still various studies exploring this trade-off [E].
5. $e=h_{L} - y^*$ should be out of the for-loop in Algorithm 1.
6. I

**Clarifications to the evaluation settings:**
1. Why are MSE loss used to train the models for classification tasks?
2. Are attacks based on MSE or cross-entropy?
3. What is the $\epsilon$ and step-size for attacks on CIFAR10/100?
4. Are the adversarial validation samples based on PGD or FGSM?

**Other suggestions:**

In Section 4, the paper introduces the concept of the feedback pathway for the first time and mentions that the feedback pathway is separated from the forward weights in biologically-plausible learning algorithms. However, don't we still need to use the forward weights when computing the second forward pass? If the feedback pathway is a central topic for the entire section, it would be helpful to introduce this concept earlier, in Section 2.



[C] Wong et al., Fast is better than free: Revisiting adversarial training

[D] Goodfellow et al., Explaining and Harnessing Adversarial Examples

[E] Li et al., The Triangular Trade-off between Robustness, Accuracy and Fairness in Deep Neural Networks: A Survey

**Strengths And Weaknesses:**

**Strengths**

1. The adversarial robustness of biological-inspired learning algorithms such as PEPITA is both important and worth investigating.
2. A variety of evaluation settings to demonstrate the potential advantage of PEPITA-trained models in terms of their adversarial robustness against perturbations.

**Weaknesses**
Since the analysis, hypotheses, and conclusions drawn in this paper heavily rely on the empirical evaluation of adversarial robustness, a thorough and accurate assessment is crucial. Despite considering a variety of evaluation **settings** (Sec. 3.2 ~ 3.5), the main weakness of the paper lies in the lack of thorough evaluation.

1. The claim is not supported by sufficient empirical evidence.
    i.  Evaluating solely based on PGD can overestimate the robustness of the model as such gradient based methods can fail. [A, B]
    ii. Despite a large number of strong attack has been proposed since PGD, most robustness evaluation include the AutoAttack algorithm which is an ensemble of attacks. [B]
    iii. The evaluation only considers a one-hidden-layer model, where the standard generalization on CIFAR-10 and CIFAR-100 is challenging. As a result, it is unclear whether any meaningful conclusions can be drawn regarding the model's robustness properties on those two datasets.
2. Presentation and writing can be improved.
    i. See requested changes.


[A] Mosbach et al., Logit Pairing Methods Can Fool Gradient-Based Attacks

[B] Croce et al., Reliable Evaluation of Adversarial Robustness with an Ensemble of Diverse Parameter-free Attacks

---

### Review · Reviewer_Xiyj · 2024-07-03

**Summary Of Contributions:**

This paper shows that the biologically-plausible learning algorithm PEPITA (Present the Error to Perturb the Input To modulate Activity) has higher intrinsic adversarial robustness with adversarial training. Furthermore, they find that PEPITA is more beneficial for the natural-vs-adversarial performance trade-off. Overall, they argue that biologically-plausible trained ANNs could be more robust against adversarial attacks because humans are not affected by adversarial attacks.

**Audience:**

Yes

**Broader Impact Concerns:**

It seems that there is no concern about ethical issues.

**Claims And Evidence:**

No

**Requested Changes:**

I think the main idea and motivation are convincing, but the experimental results are insufficient to generalize their argument. Please see weakness section above.

**Strengths And Weaknesses:**

Strength
* They provide insight into the capacity of biologically-plausible learning algorithms on adversarial robustness.

Weakness
* I agree that exploring adversarial robustness is important, and the attempt to connect it with training methods (e.g., backpropagation, biologically-plausible training methods) is meaningful. However, the authors selected only one method among various biologically-plausible training methods. This makes the contribution of the paper weak, as mentioned in the limitations and future work. I believe it needs at least one more method to argue that biologically plausible trained ANNs might be robust against adversarial attacks. Alternatively, it would be better if the authors show why the PEPITA training mechanism is better than BP for adversarial training theoretically. Or, the authors could demonstrate that PEPITA is a general/representative method of biologically plausible trained ANNs theoretically/empirically for generalizing their experimental results.
* It would be better if the authors add experimental results on various adversarial attack methods such as AutoAttack [1] and C&W [2]. The results are shown with only one adversarial attack method, so it is hard to generalize the findings.

[1] Reliable evaluation of adversarial robustness with an ensemble of diverse parameter-free attacks, ICML 2020

[2] Towards Evaluating the Robustness of Neural Networks, IEEE Symposium on Security and Privacy 2017

---

### Decision · Action_Editor_g1iT · 2024-08-03

**Recommendation:** Reject

**Comment:**

While the finding that biologically-plausible training is more robust than backpropagation is interesting, the papers lack comprehensive empirical verifications. It would be better to consider more adversarial attacks and more training methods to get a convincing finding. Furthermore, the authors do not provide a feedback to the reviewers.

**Audience:**

Yes, adversarial robustness is an important issue in neural network training. It is interesting to see the connection between adversarial  robustness and the training algorithms.

**Claims And Evidence:**

The paper shows that biologically-plausible training is robust to adversarial tackles. This claim is verified on experimental comparison with backpropagation. However, reviewers have some issues regarding the empirical evidence on the claim. Especially, the empirical verification only considers PGD attacks and a specific biologically-plausible training algorithm (PEPITA). These make the empirical verification not quite convincing and it is not clear whether the findings in the paper can be generalized to other more popular attacks and other biologically-plausible training algorithms.